# Limited Supply of Protein and Lysine Is Prevalent among the Poorest Households in Malawi and Exacerbated by Low Protein Quality

**DOI:** 10.3390/nu14122430

**Published:** 2022-06-11

**Authors:** Molly Muleya, Kevin Tang, Martin R. Broadley, Andrew M. Salter, Edward J. M. Joy

**Affiliations:** 1School of Biosciences, Future Food Beacon, University of Nottingham, Sutton Bonington Campus, Loughborough, Leicestershire LE12 5RD, UK; martin.broadley@nottingham.ac.uk (M.R.B.); andrew.salter@nottingham.ac.uk (A.M.S.); 2Faculty of Epidemiology and Population Health, London School of Hygiene & Tropical Medicine, Keppel Street, London WC1E 7HT, UK; kevin.tang1@lshtm.ac.uk (K.T.); edward.joy@lshtm.ac.uk (E.J.M.J.); 3Rothamsted Research, West Common, Harpenden, Hertfordshire AL5 2JQ, UK

**Keywords:** protein quality, amino acids, digestibility, protein deficiency, household survey

## Abstract

We estimated dietary supplies of total and available protein and indispensable amino acids (IAAs) and predicted the risk of deficiency in Malawi using Household Consumption and Expenditure Survey data. More than half of dietary protein was derived from cereal crops, while animal products provided only 11%. The supply of IAAs followed similar patterns to that of total proteins. In general, median protein and IAA supplies were reduced by approximately 17% after accounting for digestibility, with higher losses evident among the poorest households. At population level, 20% of households were at risk of protein deficiency due to inadequate available protein supplies. Of concern was lysine supply, which was inadequate for 33% of households at the population level and for the majority of the poorest households. The adoption of quality protein maize (QPM) has the potential to reduce the risk of protein and lysine deficiency in the most vulnerable households by up to 12% and 21%, respectively.

## 1. Introduction

Protein containing appropriate amounts of indispensable amino acids (IAAs) is an essential component of the human diet. Protein derived from animal products is generally regarded as high quality due to its IAA content and digestibility, and its consumption is projected to increase in low-income countries as socioeconomic status improves and the global population continues to rise [1] However, livestock production raises environmental issues due to high greenhouse gas (GHG) emissions, nitrogen pollution, and increased land degradation, mainly due to the production of crops required for animal feed [2,3,4]. In addition, excessive consumption of red meat is associated with an increased risk of developing cardiovascular diseases and some forms of cancer [5]. Thus, shifting diets towards more plant-based protein is often promoted on environmental sustainability and human health grounds [6]. However, in low-income countries where plant-based diets already predominate, greater consumption of animal-source foods may be required to help alleviate chronic malnutrition and micronutrient deficiencies [7,8], and associated GHG emissions may rise [9,10].

Animal and plant protein differ not only in their IAA composition but also in their digestibility, with the latter being of lower quality due to the presence of antinutrients such as phytates, tannins, trypsin, and protease inhibitors [8,11]. The digestibility of protein can be as low as 56% in some plant foods such as potatoes, compared with >85% for most animal-source foods [12]. The low protein quality of a typical plant-based diet consumed in low-income regions of the world is estimated to be the major driver of protein and IAA inadequacies. Studies using Food and Agriculture Organization (FAO) Food Balance Sheets, which report food available for human consumption at the national level, and dietary surveys indicate a high risk of deficiency of protein, particularly lysine, due to inadequate dietary supplies in low-income countries, particularly in Southeast Asia and sub-Saharan Africa where plant-based diets dominated by cereals and starchy roots are consumed [7,13,14,15]. 

According to Ghosh et al. [14], the prevalence of protein inadequacy calculated after correcting for protein quality is significantly higher than when total protein is used and can range from 20 to 90% for some of the poorest countries. Of particular concern is the strong association between available or digestible protein and IAA supply at national levels, with stunting levels in children under the age of 5 [7,14,16]. This was highlighted by Semba et al. [17], who showed a significantly lower concentration of circulating amino acids in the blood plasma of stunted children compared with non-stunted children in Malawi. The fortification of wheat flour with the amino acid lysine was found to have positive effects on diarrheal morbidity and on several nutritional and immunological biomarkers related to protein in Chinese and Syrian children, men and women [18,19]. This was attributed to improved protein utilization and a possible direct effect of lysine in the gastrointestinal tract.

While it is accepted that a largely cereal-based diet is associated with a high risk of protein and lysine deficiency, the proportion of such sources in the diet at which risk of inadequacy becomes apparent remain unclear. Arsenault and Brown [20] found that protein quality did not have a large impact on the overall prevalence of inadequate protein intake in a cohort of children aged 6–35 months from low-income countries who consumed a predominantly plant-based diet, including either breast milk or cow’s milk. Although the percentage of animal protein in the diet was not clear, it can be inferred that a largely plant-based diet can provide sufficient protein and IAAs under certain circumstances. A study by De Gavelle et al. [21] reported that dietary protein quality becomes crucial when the proportion of plant protein exceeds 70%. 

In 2013, the FAO recommended that protein quality be determined according to the Digestible Indispensable Amino Acid Score (DIAAS), which assesses quality according to the true ileal digestibility of each individual IAA. Although the DIAAS is now the recommended measure of protein quality, its use and adoption in practice rely on the development of a true ileal amino acid digestibility database of foods, determined preferably in humans, growing pigs, or rats, in that order. Data on the true protein and IAA ileal digestibility of foods are limited but perhaps now sufficient for evaluating the available or digestible protein and IAA supplies of many diets [22].

The objectives of this study were to characterize dietary protein supplies and deficiency risks in Malawi and to assess the impact of protein quality, corrected using ileal digestibility values, on the adequacy of dietary protein and IAA supplies across different subsections of the population. Using data from recent rounds of the Malawi-based national Integrated Household Survey (IHS), previous studies have characterized dietary micronutrient supplies and deficiency risk and modelled the effects of different interventions on alleviating deficiencies, finding important differences between socioeconomic groups [23,24,25]. Here, we use a similar approach to estimate the potential effects on dietary protein and lysine intakes of introducing the nutritionally enhanced quality protein maize (QPM) into the Malawian diet. 

## 2. Methods

### 2.1. Estimates of Total Protein and Indispensable Amino Acid Composition

The household consumption and expenditure survey (HCES) data were obtained from the Malawi Fourth Integrated Household Survey (IHS4), conducted between April 2016 and April 2017 by the National Statistics Office of Malawi in partnership with the World Bank’s Living Standards Measurement Study (LSMS) [26]. The HCES is routinely administered every 5 years, and the IHS4 covered 779 enumeration areas (EAs) in Malawi that comprised a nationally representative sample of 12,447 households. The IHS4 comprised of a single-visit 24-module questionnaire covering data on household consumption, living standards, expenditures, and other measures of social and economic welfare. In the food consumption module of the household questionnaire, participants were requested to recall the food consumed in the household over the past 7 days from a standardized list of 136 food items that are typically consumed in the Malawian diet. A comprehensive description of how nonstandard metrics recorded in the survey were transformed into standard and usable metrics is reported by Tang et al. [25]. 

The food items were matched for total protein, moisture, energy, and IAA content using Food Composition Tables (FCTs) (Appendix A). For each food item, the composition data for protein, energy, and moisture (g/100 g edible portion (EP)) were obtained from a single FCT. Approximately 72% of these values were obtained from the Malawi FCT [27], while 26% were obtained from the South Africa FCT [28]. Food items from the South Africa FCT consisted predominantly of processed products such as biscuits, infant cereals, formulas, and beverages that were not found in the Malawi FCT. Less than 3% of food items were obtained from the West Africa FCT [29] or the USDA FoodData Central Database [30]. For foods that were ambiguously defined in the survey, the authors used local knowledge and literature to match appropriate items. For example, for a food item listed as “fish” in the IHS4, compositional data for tilapia fish (Chambo in local language) were selected as one of the most commonly consumed types of fish in Malawi. Where single matches were not deemed appropriate, several food matches were identified, and an arithmetic mean was calculated. For example, for a food item listed as “mushroom” in the IHS4, an average value of two local mushroom varieties in the Malawi FCT was determined. Amino acid data were obtained almost solely from the USDA FoodData Central Database due to very limited availability from other more geographically relevant sources. For cases where the sum of IAA composition differed by >10% with total protein content, amino acid content was scaled to total protein content. This recalculation was done for >80% of the food items. Scientific articles were used to assign amino acid data to a few food items that were missing in the USDA FoodData Central Database, e.g., infant formulas and edible insects. 

### 2.2. Estimates of Available Protein and Indispensable Amino Acid Composition

An in-house protein and IAA digestibility database was created in which true or standardized ileal protein or IAA digestibility data were compiled from scientific articles [31]. A protein or IAA digestibility coefficient was assigned to all food items, after which available or digestible protein and IAA compositions (g/100 g EP) were calculated after applying the respective digestibility coefficients to the total protein and IAA values. For food items with missing digestibility data, an estimation was made using the closest match in the same food group. For example, in the legume and nuts food group, the digestibility of pigeon pea was estimated using values for common bean; in the roots and tubers food group, the digestibility of coco yam was estimated using digestibility values for potato. Digestibility values for processed products were estimated using the best match of the major ingredient; for example, the digestibility of pasta was estimated using digestibility values of wheat, and the digestibility of popcorn was estimated using values for maize or corn. Since data on the digestibility of fruits and vegetables are not available, a mean digestibility coefficient for each of the plant-based food items in the survey was calculated and used as an estimate. Digestibility was not corrected for fats and oils because of their minimal contributions to dietary protein and IAA intake. 

### 2.3. Estimates of Protein and Indispensable Amino Acid Supplies and Prevalence of Sub-Optimal Supplies

Food composition data, i.e., total and available protein and IAA (g kg^−1^, DW EP), were integrated with food consumption data (kg household^−1^ day^−1^, DW EP) to calculate the supply of nutrients at the household level. The total and available supply of nutrients for each household were expressed per adult male equivalent (AME) based on the household demographic composition and assuming that food is distributed according to the energy requirements of the individuals within the household [32]. An energy requirement of 2000 kcal day^−1^, adult male weight of 65 kg, and a moderate physical activity level (PAL) of 1.6 were assumed. The prevalence of inadequate protein and IAA supplies were estimated by comparing household supply (g AME^−1^ day^−1^) with the estimated average requirement (EAR) of an adult male (Appendix A) with an average body weight of 65 kg [33].The EAR is defined as the nutrient intake level that meets the needs of 50% of an age- and sex-specific population [34]. 

The impact of protein quality was assessed by comparing the prevalence of inadequate supplies based on total and available supply. In addition to the data being stratified according to rural/urban residences and administrative region (northern, central and southern), data were also stratified according to the socioeconomic position (SEP) of the household; households were classified into wealth quintiles on the basis of total annual-adjusted household expenditure per capita, with SEP 1 being the poorest and SEP 5 the wealthiest. The supplies of protein and IAA and the prevalence of suboptimal supplies were also compared across the five SEPs. The contributions of food groups to protein and IAA were quantified by assigning food items to the relevant food groups, i.e., cereals, legumes and nuts, animal products, vegetables, roots and tubers, dairy, fruits, and others. ‘Others’ included food groups with minor contribution (less than 1%) to protein and IAA supply such as fats and oils, sugars, spices and condiments. For a comprehensive insight into the consumption of animal products, this food group was further split into the following categories: poultry including eggs, red meat, and fish. The food groups supplying protein and IAAs were also assessed according to the household SEP.

### 2.4. Simulating the Addition of Quality Protein Maize to the Diet on Supplies and Prevalence of Sub-Optimal Supplies

The effectiveness of introducing quality protein maize (QPM) to the Malawi diet to reduce the prevalence of inadequate supplies was simulated by substituting the total and available protein and IAA compositions of currently used maize with those of a typical QPM variety using estimates derived from Prasanna et al. [35] and Yin et al. [36]. This was applied to the main IHS4 food items comprising maize as the major food ingredient i.e., ‘maize *ufa mgaiwa* (normal flour)’, ‘maize *ufa* refined (fine flour)’, ‘maize *ufa madeya* (bran flour)’, ‘maize grain (not as *ufa*)’, and ‘green maize’. The total protein content of QPM is not expected to be significantly different from that of currently used maize varieties; rather the amino acid profile of QPM is modified. However, the digestibility of QPM is greater than that of currently used maize varieties, and as such, a comparison between the available protein supplies of the current scenario versus the QPM scenario was made. In terms of IAA composition, for the QPM scenario, a factor of 1.3 was applied over the current scenario to lysine composition. The QPM scenario food composition data were analyzed as described in the previous sections. 

## 3. Results

### 3.1. Socioeconomic and Demographic Information of Households 

Table 1 shows the demographic information of households that participated in the IHS4. A total of 12,447 randomly sampled households from the three administrative regions of Malawi (northern, central and southern) were interviewed, with 82% living in rural areas and 18% in urban areas. The number of households sampled in each region generally followed the population distribution of Malawi, with more people living in the rural areas compared with the urban areas and more people living in the southern region compared to the northern and central regions. For each region, the same proportion of households belonging to the different SEP was interviewed so that each category of SEP was equally represented. In total, each SEP comprised approximately 20% of households in the survey (Table 1). A detailed summary of the survey population is given by Tang et al. [25]. 

### 3.2. Food Groups Supplying Protein and Indispensable Amino Acids to the Malawian Diet

Cereal was the major food group supplying total and available protein to the Malawian diet (Figure 1A), with a mean supply of 66%, followed by legumes and nuts (15%), animal products (11%) and vegetables (4%), with other food groups contributing minor proportions. The proportion of food groups supplying protein was similar when comparing total and available supply, although there was a slight increase in the proportion of supply from animal products based on available supply. Figure 1B shows the proportions of food groups supplying protein according to household SEP. As household SEP increased, supply from cereals decreased from 78% for SEP 1 (poorest) to 54% for SEP 5 (wealthiest). The opposite was true for the proportion of supply from animal products, which increased from 5% for SEP 1 to 19% for SEP 5, and legumes and nuts, which increased from 11% for SEP 1 to 17% for SEP 5. The proportion of supply from other food sources such as vegetables, roots, and tubers, was similar across all five categories of SEP. Animal products and vegetables were equally important for the poorest households (SEP 1) as they contributed similar proportions to the protein supply (ca. 4%). The proportion of food groups supplying the majority of IAAs to the diet followed a similar trend (Appendix A). However, the supply of lysine and tryptophan from cereals was lower compared with protein, i.e., 44% and 55% respectively, vs. about 66% for protein, while the supplies of leucine and sulfur amino acids (SAA) from cereals were higher, i.e., 76% and 71%, respectively. Lysine supply showed the greatest deviation from that of protein, with legumes, nuts, and animal products supplying similar proportions (~23%). Animal products were further classified into poultry (including eggs), red meat, fish, and dairy to obtain comprehensive insights into the animal source foods most commonly consumed (Appendix A). The most consumed animal source food in the Malawian diet was fish, which accounted for more than 50% of the animal protein consumed. Fish was the most important animal source food for the lowest SEP, while supplies of fish, red meat, and poultry were similar in the households of the highest SEP. 

### 3.3. Total and Available Protein and Indispensable Amino Acids Supply 

After accounting for digestibility, approximately 17% of proteins were not available. Losses due to digestibility ranged between 9 and 21% for individual IAAs. The median total protein supply was 80 g AME^−1^ day^−1^, and this was reduced to 65 g AME^−1^ day^−1^ after accounting for digestibility (Table 2). When data were disaggregated according to SEP, the median total supply ranged from 48 g AME^−1^ day^−1^ for the poorest households to 120 g AME^−1^ day^−1^ for the wealthiest households. The median protein supply was reduced by 19% for households in SEP 1, compared with 16% for households in SEP 5 after accounting for digestibility. When QPM was substituted for the conventional maize varieties, losses of protein due to digestibility were reduced from 19 to 13%, and available protein supply per AME increased by 8% at the population level and by 10% for the poorest households (Table 2). The median supply of all IAAs is given in Appendix A, while that of lysine is shown in Table 3. Similar to protein, the trend in the supply of IAA followed the same pattern as that of protein, i.e., at least twofold greater supply for the wealthiest households compared with the poorest households. The median supply of lysine showed the largest deviation, with a greater reduction in supply due to digestibility in the poorest households (22%) compared with the wealthiest households (14%). If QPM is introduced into the Malawian diet, total and available lysine supply are projected to increase by 16–20%. Notably, the increase in supply was greatest for the poorest households (Table 3). 

### 3.4. The Risk of Protein and Indispensable Amino Acid Deficiencies Due to Suboptimal Supplies

The supplies of protein and IAAs (g AME^−1^day^−1^) were compared with the EAR of an adult male with an average weight of 65 kg to determine the adequacy of supplies (Appendix A). At population level, and based on total protein supply, 12% of the population was at risk of deficiency due to inadequate dietary supplies (Table 4). When considering available protein supply, the risk of deficiency increased to 21%, meaning that at the population level, improving protein digestibility could reduce the risk of protein deficiency due to inadequate supply by up to 9%. The risk of deficiency increased according to SEP, with >50% of households in SEP 1 at risk of protein deficiency based on available supply. The importance of protein digestibility also increased as household wealth decreased, i.e., improving protein digestibility could reduce the risk of deficiency by only 1% for SEP 5 (wealthiest) compared with 19% for SEP 1 (poorest). The introduction of QPM into the Malawian diet has the potential to reduce the risk of protein deficiency from 21% to 17% at the population level, while for the most vulnerable households, risk of protein deficiency can be reduced from 58% to 51% (Table 4).

Pertaining to IAAs, the risk of deficiency was lower than that for protein except for lysine (Appendix A). Approximately one third of the population was at risk of lysine deficiency based on available supplies (Table 5), while the majority of households in SEP 1 were at risk of lysine deficiency due to inadequate supplies. Similar to protein, digestibility was most important for the poorest households, as improving lysine digestibility could potentially reduce risk by up to 18 and 22% for SEP 1 and 2 respectively, compared with only 2% for SEP 5. If QPM is introduced into the Malawian diet, the risk of lysine deficiency can potentially be reduced by 12% at the population level and up to 21% for the poorest households (Table 5). 

## 4. Discussion

Dietary protein and IAA supplies in Malawi varied widely and were largely associated with household wealth. Among other demographic characteristics such as geographic location and whether a household was located in a rural or urban area, household SEP was the most important attribute explaining the large variation in supplies. This is consistent with other studies in which protein and IAA supplies have been found to be strongly correlated with the gross domestic product (GDP) at country level and with household wealth when dietary surveys were used [7,14]. According to Grigg [37], “the map of protein consumption is the map of the level of economic development,” and while this is evident when comparing developed and developing countries, this variability also exists among households of different SEP within a country. Indeed, high protein animal source foods are at least two times more expensive per unit protein than the energy-dense but low protein plant-based foods such as cereals and legumes [38]. In addition to the cost, the composition and quality of animal protein is superior to that of plant protein. This explain why, not only the supplies of protein and IAA was increased as household SEP increased, but also the pattern of foods providing protein also reflected the household wealth, with lower consumption of animal protein among the poorest households. 

The risk of protein and lysine deficiency was high and exacerbated by poor protein quality especially among the poorest households. In general, variations in the supplies of protein and IAA were similar except for lysine, with a deviation of up to 77% from the median supply compared with protein with a deviation of up to 50%. One important factor that influences the quantity and source of protein supply in a given country is the local environmental conditions, which determine the choice of staple crop [37]. In Malawi, maize is the staple crop [39,40], and the variation in the supplies of IAAs and the subsequent risk of deficiency reflect the IAA profile of maize. For example, maize is a good source of several IAAs such as leucine, aromatic amino acids (AAA), and histidine, and this explains their lower risk of deficiency (<5%) compared with protein (21%) when considering the available supplies. On the other hand, maize is limited in lysine, as indicated by the higher risk of deficiency (33%) than that for protein. In fact, the risk of deficiency of IAAs except lysine is substantially lower (3–11%) than that for protein, suggesting that for diverse diets, protein supplies alone provide a good indication of both the protein and the IAA adequacy of a diet. As supplies were largely associated with household SEP, it is not surprising that the majority of households in the lowest SEP were at risk of both protein and lysine deficiency. 

The impact of protein quality on protein and IAA supplies was more important for the poorest households than the wealthiest households. Protein quality is a measure of the digestibility and IAA profile of a protein [41]. The comparisons between total and available protein and IAA supplies show the impacts of poor digestibility, causing about 17% of protein and IAA losses across households, with the greatest nutritional significance for households in the lowest SEP. The losses in protein and IAA supplies also reflect the digestibility of the staple cereal maize, which was estimated to be about 82%. It therefore can be suggested that the risk of protein and IAA deficiencies due to inadequate supplies may be greater in regions where the protein digestibility of the staple crops is lower, particularly where diets are based on sorghum and millet varieties with digestibility of around 66–79% [31]. The effect of the other component of protein quality, which is IAA profile, is demonstrated in the greater risk of lysine deficiency than protein as mentioned previously. The risk of lysine deficiency increased exponentially according to SEP and was substantial even for the moderately wealthy households (i.e., SEP 3). Protein quality was critical at cereal protein proportions of >70% as indicated by the large increase in the risk of lysine deficiency from SEP 3 to 5. For cereal protein proportions of around 50%, the quantity of protein consumed was more important, as shown in the diets of the wealthiest households, with protein supply even exceeding requirements by a large magnitude. In agreement with De Gavelle et al. [21], the protein and IAA adequacy of a largely plant-based diet is dependent mainly upon dietary diversity. Diets with protein derived from >70% plant protein including choices of legumes, nuts, and seeds were estimated to provide sufficient protein and lysine due to the complementarity of the different food sources [21]. For example, legumes can complement the shortage of lysine in cereals, while cereals complement SAAs, which are the first limiting amino acids in legumes. 

However, the main factor hindering the supply of diverse diets to vulnerable populations is poverty. In Malawi, poorer households are more likely to live in rural areas, with greater distances to markets and roads and less engagement in wage employment [25]. They are also likely to derive less benefit from large-scale food fortification programmes, including of maize flour and cooking oil, since they typically purchase processed foods less often and in smaller quantities [25,42]. In that regard, an intervention to biofortify staple crops may present a more accessible and sustainable option as vulnerable population groups are likely to be small-holder subsistence farmers. Quality protein maize (QPM) is a group of biofortified maize varieties with elevated levels of lysine and tryptophan and with crop development led by the International Maize and Wheat Improvement Center (CIMMYT) [35]. It is estimated that QPM has protein digestibility almost similar to that of casein and a balanced amino acid profile, with at least 30% more lysine compared than traditional maize varieties [35,39]. In this study, the introduction of QPM with ~10% improvement in digestibility (91% vs. 82% for maize) and a modest 30% enhancement of lysine content was simulated. The introduction of QPM could potentially augment both protein and lysine supplies and substantially reduce the risk of deficiency among vulnerable households by up to 21%. Intervention with QPM has been found to have positive effects on child growth in randomized controlled trials conducted in several developing countries [43,44]. However, QPM has not been widely adopted in sub-Saharan Africa compared with newer biofortified crop varieties, such as pro-vitamin A maize, despite its clear nutritional benefits. Nyakurwa et al. [40] highlighted some constraints to its adoption that still need to be overcome including a lack of farmer awareness and supportive government policies. Like other staple biofortified crops, government–private sector partnerships and a robust communication strategy in nutrition education campaigns are key to encourage adoption [25,40]. 

Although the risk of lysine deficiency can be significantly reduced if QPM is adopted in the diet, the risk remains substantial among households in the lowest SEP (up to 21% at population level and 64% for SEP 1). The risk of lysine deficiency could be reduced further if lysine enhancement levels are greater than the conservative 30% increase used in this study. The proportional increase in lysine concentrations of QPM relative to traditional maize varieties can vary greatly, ranging from 10 to 58% for some Canadian adapted cultivars [45] and from 30 to 82% for Ethiopian adapted QPM [46]. Therefore, it will be important to understand the levels of lysine enhancement that are likely to be achieved in QPM varieties adapted to the typical environmental and agronomic conditions in Malawi. While intervention with QPM can have a substantial impact on protein and IAA supplies, it is clearly not sufficient on its own and needs to be complemented with other interventions. An increase in the supply of animal source foods is especially important because the high risk of protein and lysine deficiency among households of the lowest SEP is also accompanied by a high risk of the deficiency of micronutrients including vitamin A and zinc [25]. As fish supply was almost equally available across all household SEPs, interventions increasing availability of this source of protein need to be explored. 

Data from the HCES food consumption modules provide an important resource for nutritional assessment and can inform nutrition policy and programming. Although adult male equivalent (AME) factors are considered a valid proxy of apparent individual nutrient intakes within a household, giving a general picture of the status of nutrient supplies in a country [47], it is limited in its ability to identify differing risks between population age and sex groups. Considering the association between protein inadequacy and undernutrition in children [7,14], the current study would be well complemented by looking at dietary intake data from children under 5 and women of reproductive age among rural households of the lowest SEP. A more targeted study could also enable the adjustment of protein requirements for energy deficits and infections, which increase protein requirements [14] and are likely to be more prevalent among households in the lowest SEP. 

## 5. Conclusions

Analysis of data from the Malawi Integrated Household Survey 2016–2017 (IHS4) predicted a high risk of protein and IAA deficiencies due to inadequate dietary supplies, particularly amongst those of the lowest socioeconomic status. High-quality protein sources from animal products were less available to these households, who derived the majority of their protein supply from maize, the staple crop. As a consequence, the further losses of protein and IAA supply due to protein quality disproportionately affected the poorest households compared with the wealthiest households. An improvement in protein digestibility could reduce the prevalence of inadequate protein and lysine supply by up to 20% for the poorest households but only 2% for the wealthiest households. The risk of lysine deficiency was substantial in the whole population, and evidence from other sources is needed, which if corroborated, suggests interventions need to be investigated rapidly with a focus on reaching the most vulnerable populations. An intervention involving the adoption of QPM has the potential to reduce the risk of inadequate protein supplies by up to 7% due to improved protein digestibility, while lysine deficiency risk could be reduced by up to 21%. 

## Figures and Tables

**Figure 1 nutrients-14-02430-f001:**
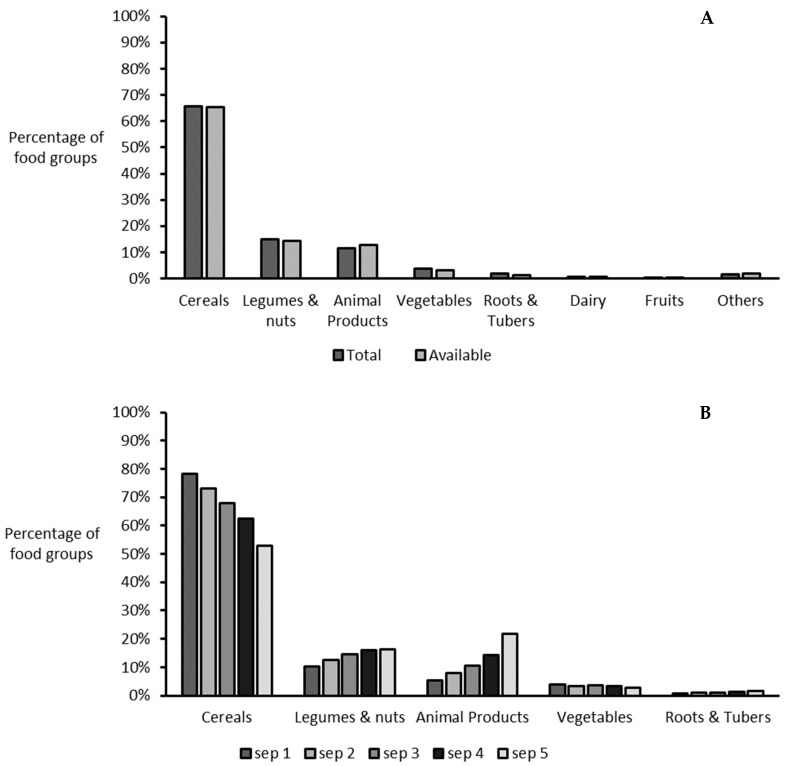
The food groups supplying total and available protein (**A**) and the top 5 food groups supplying total protein to the Malawian diet by household socioeconomic position (**B**). SEP: socioeconomic position (1 = poorest quintile, 5 = wealthiest quintile).

**Table 1 nutrients-14-02430-t001:** The numbers (and percentages) of the households interviewed in IHS4 (2016–2017) by socioeconomic position and region.

Region	SEP 1	SEP 2	SEP 3	SEP 4	SEP 5	Total
Northern	399 (16%)	492 (20%)	532 (21%)	525 (21%)	543 (22%)	2491 (20%)
Central	734 (17%)	827 (20%)	862 (20%)	902 (21%)	895 (21%)	4220 (34%)
Southern	1357 (24%)	1170 (20%)	1096 (19%)	1062 (19%)	1051 (18%	5736 (46%)
Total	2490 (20%)	2489 (20%)	2490 (20%)	2489 (20%)	2489 (20%)	12,447

SEP: socioeconomic position (1 = poorest quintile, 5 = wealthiest quintile).

**Table 2 nutrients-14-02430-t002:** The median total and available protein supplies (g AME^−1^ day^−1^) by household socioeconomic position, including under a scenario with universal adoption of quality protein maize (QPM).

SEP	Total Protein	Available Protein	Available Protein QPM-Scenario
1	48	39	43
2	67	55	60
3	81	67	72
4	97	79	85
5	120	101	106
ALL	80	65	70

SEP: socioeconomic position (1 = poorest quintile, 5 = wealthiest quintile), AME: adult male equivalent, QPM: quality protein maize.

**Table 3 nutrients-14-02430-t003:** The median total and available lysine supplies (g AME^−1^ day^−1^) by household socioeconomic position, including under a scenario with universal adoption of quality protein maize (QPM).

SEP	Current Scenario		QPM Scenario	
	Total Supply	Available Supply	Total Supply	Available Supply
1	1.7	1.3	2.0	1.7
2	2.5	2.0	2.9	2.5
3	3.2	2.6	3.6	3.1
4	3.9	3.3	4.5	3.8
5	5.5	4.7	6.0	5.3
ALL	3.1	2.5	3.6	3.0

SEP: socioeconomic position (1 = poorest quintile, 5 = wealthiest quintile), AME: adult male equivalent, QPM: quality protein maize.

**Table 4 nutrients-14-02430-t004:** The percentages of households at risk of protein deficiency due to inadequate dietary supply, including under a scenario with universal adoption of quality protein maize (QPM), stratified by socioeconomic position.

SEP	Current Scenario			QPM Scenario		
	Total Supply	Available Supply	Difference	Total Supply *	Available Supply	Difference
1	39	58	19	39	51	12
2	12	24	12	12	19	7
3	6	13	7	6	10	4
4	3	6	3	3	5	2
5	1	2	1	1	2	1
ALL	12	21	9	12	17	5

SEP: socioeconomic position (1 = poorest quintile, 5 = wealthiest quintile), QPM: quality protein maize.* Total protein supply under the QPM scenario is similar to current scenario as gross protein composition of QPM is not significantly different from that of traditional maize varieties.

**Table 5 nutrients-14-02430-t005:** The percentages of households at risk of lysine deficiency due to inadequate dietary supply including under a scenario with universal adoption of quality protein maize (QPM) stratified by socioeconomic position.

SEP	Current Scenario			QPM Scenario		
	Total Supply	Available Supply	Difference	Total Supply	Available Supply	Difference
1	63	82	19	47	64	17
2	25	47	22	14	26	12
3	10	23	13	6	11	5
4	4	9	5	3	5	2
5	1	3	2	1	1	0
ALL	21	33	12	14	21	7

SEP: socioeconomic position (1 = poorest quintile, 5 = wealthiest quintile), QPM: quality protein maize.

## Data Availability

Data is contained within the article or Appendix A. Dataset on the ileal protein and amino acid digestibility of foods generated in this study can be found at: https://data.mendeley.com/datasets/gz3cx7d5f4/1—DOI:10.17632/gz3cx7d5f4.1.

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
