# Peer review of "Limited Supply of Protein and Lysine Is Prevalent among the Poorest Households in Malawi and Exacerbated by Low Protein Quality"

_nutrients, 2022, doi:10.3390/nu14122430_

Round 1

Reviewer 1 Report

The paper is very interesting and very well written. Protein deficiency is a highly impacting issue, especially in low-income countries. The manuscript largely demonstrates that protein deficiency is related to socioeconomic status and identifies possible policy interventions mitigating the problem occurrence.

So, I have just a few suggestions for the introduction.  

The first one is to add a citation at line 40, particularly Tuomisto HL. The complexity of sustainable diets. Nat Ecol Evol. 2019 May;3(5):720-721. DOI: 10.1038/s41559-019-0875-5. PMID: 30988495.

The second one is to add a few words about the consequence of lysine deficiency, especially considering the special issue (line 51).

 Moreover, two small typos:

Line 405 please indicate the extended AME as it was not previously introduced;

Tables (all): please add a footnote including the extended QPM.

Author Response

Thank you very much for taking the time to review our paper. Please find below our response to the questions raised.

Point 1: The first one is to add a citation at line 40, particularly Tuomisto HL. The complexity of sustainable diets. Nat Ecol Evol. 2019 May;3(5):720-721. DOI: 10.1038/s41559-019-0875-5. PMID: 30988495.

Response 1: This reference has now been added on line 41 of the new manuscript. 

Point 2: The second one is to add a few words about the consequence of lysine deficiency, especially considering the special issue (line 51).

Response 2: The main function of lysine in the human body is for protein synthesis as such implications of lysine deficiency can lead to non-specific signs of protein deficiency. On line 62-66 of the new manuscript, we have added a comment based on evidence from some studies indicating an improvement in diarrheal morbidity, nutritional and immunological markers related to protein in Chinese and Syrian children, men and women. This was attributed to improved protein utilization and a possible role of lysine in modulating gastrocolonic and small intestinal transit. 

Point 3: Moreover, two small typos:

Line 405 please indicate the extended AME as it was not previously introduced; 

Tables (all): please add a footnote including the extended QPM.

Response 3: The abbreviation for the adult male equivalent (AME) was first introduced in the Methods section, line 156 of new manuscript. However, this has been repeated in the Discussion on line 414 new manuscript as recommended by the reviewer. 

The abbreviations for AME and QPM are now defined in the footnotes of the relevant tables i.e., AME for Tables 2 and 3 and AME and QPM for Tables 4 and 5. 

Reviewer 2 Report

It is better to extending the result with protein in different food items, including  cereal type and diffrernt animal foods.

Author Response

Point 1: It is better to extending the result with protein in different food items, including  cereal type and diffrernt animal foods.

Response 1: We thank the reviewer for the comment and for taking the time to review our manuscript. We are not sure if we completely understand the question, considering the section being indicated is not specified, but we believe the reviewer may be recommending that, instead of showing protein supplies disaggregated by food groups (Fig 1A and B), we should list supplies based on specific food items. 

We respectfully disagree with this because the disaggregation of the 136 food items in the survey into food groups (Method section, line 108) is the best way to reduce this data in the most meaningful and clear way to communicate the results. This is also unnecessary considering that the intervention of interest is quality protein maize (QPM) and not broader dietary diversity initiatives. This intervention was chosen as maize is the major source of protein for ALL households and even more important for households in the lowest socioeconomic positions who are most vulnerable to protein and lysine deficiency. In terms of animal products, a further disaggregation of the animal products into poultry, red meat, fish and dairy was done and this data is shown in Fig S1, Supplementary data 2. Supplementary data 1 shows all the foods that were listed in the survey along with their total/available protein and amino acid composition. We believe this allows readers who may need more information on the type of food items typically consumed in Malawi to access the appropriate information.